## [Review Process File · Molecular Systems Biology]

Title: Cytometry-based single cell analysis of intact epithelial signaling reveals MAPK activation divergent from TNF- α -induced apoptosis *in vivo*

Authors: Alan J. Simmons^{1,2*}, Amrita Banerjee^{1,2*}, Eliot T. McKinley^{1,3}, Cherie' R. Scurrah^{1,2}, Charles A. Herring^{1,4}, Leslie S. Gewin³, Ryota Masuzaki⁵, Seth J. Karp⁵, Jeffrey L. Franklin^{1,2}, Michael J. Gerdes⁶, Jonathan M. Irish⁷, Robert J. Coffey^{1,2,3,8}, Ken S. Lau^{1,2,4}

¹ Epithelial Biology Center, Vanderbilt University Medical Center, Nashville, TN

² Department of Cell and Developmental Biology, Vanderbilt University Medical Center, Nashville, TN

³ Department of Medicine, Vanderbilt University Medical Center, Nashville, TN

⁴ Department of Chemical and Physical Biology, Vanderbilt University Medical Center, Nashville, TN

⁵ The Transplant Center and Department of Surgery, Vanderbilt University Medical Center, Nashville, TN

⁶ Life Sciences Division, GE Global Research, Niskayuna, NY

⁷ Departments of Cancer Biology, Pathology, Microbiology and Immunology, Vanderbilt University Medical Center, Nashville, TN

⁸ Department of Veterans Affairs Medical Center, Nashville, TN

Corresponding author: Ken S. Lau
Epithelial Biology Center
Vanderbilt University Medical Center
2213 Garland Ave
10475 MRB IV
Nashville, TN 37232-0441
Email: ken.s.lau@vanderbilt.edu

* Equal author contributions

Abstract

Understanding heterogeneous cellular behaviors in a complex tissue requires the evaluation of signaling networks at single cell resolution. However, probing signaling in epithelial tissues using cytometry-based single cell analysis has been confounded by the necessity of single cell dissociation, where disrupting cell-to-cell connections inherently perturbs native cell signaling states. Here, we demonstrate a novel strategy (Disaggregation for Intracellular Signaling in Single Epithelial Cells from Tissue – DISSECT) that preserves native signaling for Cytometry Time-of-Flight (CyTOF) and fluorescent flow cytometry applications. A 21-plex CyTOF analysis encompassing core signaling and cell-identity markers was performed on the small intestinal epithelium after systemic tumor necrosis factor-alpha (TNF- α) stimulation. Unsupervised and supervised analyses robustly selected signaling features that identify a unique subset of epithelial cells that are sensitized to TNF- α -induced apoptosis in the seemingly homogeneous enterocyte population. Specifically, p-ERK and apoptosis are divergently regulated in neighboring enterocytes within the epithelium, suggesting a mechanism of contact-dependent survival. Our novel single cell approach can broadly be applied, using both CyTOF and multi-parameter flow cytometry, for investigating normal and diseased cell states in a wide range of epithelial tissues.

Introduction

Characterization of protein signaling networks for systems-level analysis of cellular behavior requires the quantification of multiple signaling pathway activities in a multiplex fashion. Previous and current studies of multi-pathway epithelial signaling rely on bulk assays that hinge on the assumption of cell homogeneity in, for example, *in vitro* cell culture systems. Although useful in revealing coarse-grain biological insights into behaviors exhibited by a majority of cells (Lau *et al*, 2013, 2012, 2011), these technologies fail to address the complexities exhibited by heterogeneous cell types *in vivo*. Flow cytometry is a tractable method for detecting and quantifying signal transduction information at single cell resolution (Krutzik *et al*, 2004; Irish *et al*, 2004). CyTOF, where the limitation of fluorescence spectral overlap is overcome by the resolution of metal-labeled reagents by mass spectrometry, allows for multiplex sampling of protein signals at a network scale and at single cell resolution (Bendall *et al*, 2014, 2011). In parallel, newly developed fluorescent dyes and compensation algorithms allow 15-20 parameters to be measured simultaneously using multicolor fluorescent flow cytometry (O'Donnell *et al*, 2013). A tremendous opportunity for single cell studies lies in expanding quantitative cytometric approaches to epithelial tissues, from which many diseases arise. A significant challenge, however, is the preparation of single cell suspensions from these tissues while maintaining intact cell signaling states. Disruption of epithelial cell junctions during cell detachment perturbs native cell signaling networks (Baum & Georgiou, 2011; Pieters *et al*, 2012) and can create experimental artifacts that overwhelm native signaling. To date, strategies to quantify epithelial protein signal transduction by cytometry approaches without confounding dissociation artifacts have not been developed.

We present a novel method, DISSECT, for preparing single cell suspensions from epithelial tissues for single cell, cytometry-based signaling analyses. We use DISSECT followed by CyTOF to characterize multiple signaling pathway responses in the murine intestinal epithelium following *in vivo* exposure to TNF- α , a pleiotropic cytokine that plays significant roles

in the pathogenesis of Inflammatory Bowel Disease (Colombel *et al*, 2010), celiac disease (Chaudhary & Ghosh, 2005), and necrotizing enterocolitis (Halpern *et al*, 2006). In the villus of the duodenum, TNF- α triggers caspase-dependent cell death, creating an epithelial barrier defect that increases exposure of nutrient and microbial antigen to the underlying immune system (Lau *et al*, 2011; Williams *et al*, 2013). Remarkably, only a fraction of villus cells undergo apoptosis, and higher levels of cell death cannot be induced by a higher TNF- α dose (Lau *et al*, 2011). The existence of heterogeneous responses provides a unique opportunity to leverage the natural variation of cells for identifying perturbations that result in desirable cellular outcomes. To decipher heterogeneous responses at single cell resolution, we first provide rigorous, quantitative validation of our single cell approach in comparison to gold-standard lysate-based methods for evaluating both cellular identity and signaling. We then use DISSECT-CyTOF to quantify 21 protein and phospho-protein analytes across core signaling pathways at single cell resolution. Quantitative modeling of single cell datasets reveals that a subset of the presumably homogeneous enterocyte population exhibits combinations of signaling responses that confer sensitivity to TNF- α -induced cell death. Our results reveal novel insights into the intricacies of *in vivo* epithelial cell populations that exhibit significant complexity when perturbed and then observed at single cell resolution. Our approach can be extended to a broad range of complex, heterogeneous epithelial tissues that can be studied via the use of either multiparameter flow cytometry or CyTOF.

Results

A novel disaggregation procedure for investigating epithelial signaling heterogeneity

Tissues *in vivo* present substantial heterogeneity at the cellular level, as exemplified by the different responses of individual cells to exogenous perturbations. We modeled heterogeneous response *in vivo* by inducing villus epithelial cell death by systemic TNF- α administration. TNF- α triggered apoptosis only in a third of duodenal villus epithelial cells over a

4-hour time course (Fig. EV1A, B). The remaining cells were not in the process of cell death, as evidenced by the full recovery of intestinal morphology 48 hours after TNF- α exposure (Fig. EV1C). Heterogeneous, TNF- α -induced apoptosis occurred intermittently throughout the length of the villus, and not only at the villus tip as observed in homeostatic cell shedding (Fig. 1A, Fig. EV1D). Furthermore, TNF- α -induced apoptosis appeared to occur solely in a subset of villus enterocytes, as cleaved caspase-3 (CC3) did not co-localize with other epithelial cell type markers (goblet- MUC2: Mucin2, tuft- DCLK1: doublecortin-like kinase 1, enteroendocrine- CHGA: Chromagranin A) (Fig. 1B, Fig. EV1D, E). However, CC3 was co-localized in cells positive for Villin, a protein of enterocyte brush borders, both within the villus epithelium (dying cells) and in the gut lumen (dead cells) (Fig. EV1F). The notion of enterocyte-specific cell death was further supported by increased goblet and tuft cell fractions over time, indicating enrichment of these cell types compared to the remaining enterocytes (Fig. EV1G, H). Although enterocyte cell death occurred heterogeneously in response to TNF- α , the sensing of TNF- α ligand by TNF receptor (TNFR) appeared uniform in these cells. TNFR1 expression was observed on the basolateral membranes of all villus epithelial cells (Fig. 1C, Fig. EV1I), and was reduced in all cells uniformly upon TNF- α stimulation, consistent with internalization of the receptor in direct response to TNF- α binding (Schütze *et al*, 2008). TNFR2 was expressed at very low levels in the villus epithelium (Fig. EV1I'), supporting previous reports of its minimal role in the villus compartment (Lau *et al*, 2011). Since TNF- α sensing appeared uniform in all villus epithelial cells, we surmise that heterogeneous TNF- α responses in enterocytes may depend upon differences in signal transduction downstream of receptor binding.

A major challenge for exploring signaling heterogeneity in epithelial tissues with cytometry-based methods is the requirement of single cell suspensions. Previous attempts to probe epithelial signaling involved stimulation experiments on single epithelial cells that were already dissociated and outside of their native contexts (Lin *et al*, 2010). To study single cell

signaling in the *in situ* epithelial context, we first tested whether a single cell disaggregation procedure used routinely for flow sorting epithelial cells (Magness *et al*, 2013) (which we referred to as “the conventional method”) preserves native signaling in single cell suspensions. Briefly, the intestinal epithelium was mechanically retrieved after the intestine was acquired, washed, and longitudinally opened. The epithelium was then digested enzymatically (~10 min) and then filtered into a single cell suspension. A standard fix-perm procedure for phospho-flow was then performed, followed by phospho-specific antibody staining and cytometry analysis (Kruzik *et al*, 2004). Quantitative immunoblotting analysis on fresh intestinal tissue lysates was used as positive control. A head-to-head assessment using the same antibodies was performed by comparing median intensities from single cell flow cytometric data to integrated intensities of bands from immunoblots, which reflect cell averages in tissue lysates. This comparison demonstrated that signal transduction induced by TNF- α was not maintained with the conventional disaggregation method, as assessed by both early (p-ERK1/2, p-C-JUN) and late (p-STAT3) signals (Fig. EV2). A previous study suggested that signaling perturbations from tryptic disaggregation can be eliminated by performing digestion in live cells at low temperatures (Abrahamsen & Lorens, 2013). We tested the effect of enzymatic digestion by performing low-yield single cell disaggregation on live tissues (Appendix Fig. S1A), using gentle mechanical dissociation without any enzymes; however, signal transduction was still not preserved (Appendix Fig. S1B). Disaggregation of an intact epithelium into single cells perturbs components of epithelial cell junctions that play many roles in signaling modulation. Such disruption in live tissue may dynamically alter signaling pathways and produce experimental artifacts.

To adapt single cell signaling analysis for epithelial tissues, we developed DISSECT, a single cell dissociation method that preserves intact signaling. After the epithelium was retrieved from the animal, it was immediately fixed to maintain cellular signaling states. The epithelium was then subjected to acetone permeabilization and antigen retrieval by a detergent solution,

followed by staining and an additional fixation step to crosslink antibodies onto their epitopes. Stained epithelium was then disaggregated into single cells enzymatically followed by gentle mechanical agitation (Fig. 1D). Retrieval of single cells and their yields were robustly verified, with cells prepared by DISSECT retaining a native columnar morphology, versus the round morphology arising from the conventional method (Appendix Fig. S2, Fig. EV3A, B). Specifically, quantitative yields of single cells from DISSECT were higher than those from the conventional approach, where cell clumping induced by methanol and pronounced adhesion of single cells to plastic ware resulted in cell loss (Fig. EV3C). We tested whether native signaling is maintained throughout the DISSECT process, again by direct comparison with “gold standard” approaches performed on the same tissues. Activation of p-C-JUN and p-STAT3 were detected at 0.5 hour and 4 hours, respectively, in single cells by immunofluorescence microscopy, mirroring intact tissue staining (Fig. 2A). Singleplex flow cytometry on prepared single cell suspensions enabled the quantification of signal transduction at single cell resolution, whose median values from single cell distributions can be compared to lysate-based quantitation (Fig. 2A'). We observed upregulation of p-C-JUN early (0.5 hour) and p-STAT3 late (2 hour) at the population level, matching previously observed dynamics of these two TNF- α -activated pathways (Lau *et al*, 2011). Furthermore, median data derived from single cells prepared using DISSECT over multiple replicates qualitatively matched immunoblotting data from lysates prepared from the same tissue (Fig. EV2), in stark contrast to single cells prepared using the conventional method. Furthermore, preservation of signals using DISSECT was not further improved by perfusing the animal beforehand with fixative, indicating that our method of tissue collection does not significantly perturb native signaling (Appendix Fig. S3). By verifying the performance of DISSECT in technical and biological replicates (Appendix Fig. S4), we conclude that our procedure is robust for maintaining native signaling during single cell disaggregation.

DISSECT allows phenotypic cell profiling of complex epithelial tissues

A potential limitation of DISSECT is the possible degradation of proteins at the cell surface, thus limiting our ability to identify cell types using cell surface markers. To ensure that the DISSECT approach can preserve cell surface antigen staining for cell-type identification, we evaluated canonical markers for leukocytes and other epithelial cell types using flow cytometry in our single cell preparations. CD45⁺ cells in the intestinal lamina propria can be readily detected and increased 2 hours after TNF- α stimulation, similar to what we observed previously for immune cell types (Appendix Fig. S5A) (Lau *et al*, 2012). Specifically, we detected different populations of villus epithelial cells using goblet (CLCA1 - Calcium-activated chloride channel regulator 1), enteroendocrine (CHGA), and tuft (DCLK1) cell markers (Fig. 2B). The proportion of differentiated cells detected in the intestine matched previous reports, with goblet cells at ~10% and increasing from the duodenum to the ileum (Rojanapo *et al*, 1980; Paulus *et al*, 1993; Imajo *et al*, 2014; van der Flier & Clevers, 2009; Wright & Alison, 1984), enteroendocrine cells at ~1% (Gunawardene *et al*, 2011; Cheng & Leblond, 1974), and tuft cells at ~1% (Gerbe *et al*, 2012). Imaging-based quantification of the same tissues also confirmed these results (Fig. 2B', B''). We further tested whether our method can detect crypt stem cells using the cell surface marker LRIG1 (Appendix Fig. S5B) (Powell *et al*, 2012). Isolation of colonic crypts followed by DISSECT and flow cytometry allowed for the identification and quantification of crypt base cells, which segregate away from Na/ATPase⁺ differentiated cells (Fatehullah *et al*, 2013) (Appendix Fig. S5C). The proportion of LRIG1⁺ cells matched what was previously reported (~30% in the colonic crypt) using the same antibody (Poulin *et al*, 2014). In addition, TNF- α -induced signaling can be detected in single cells isolated from colonic crypts (Appendix Fig. S5D), as well as from colonic tumors (Appendix Fig. S5E) using flow cytometry following DISSECT. To test the general applicability of DISSECT in other epithelial tissues, we induced proliferation in the collecting ducts of the kidney and hepatocytes in the liver using an unilateral ureteral obstruction

(UUO) model and a partial hepatectomy model, respectively. GFP+ cells from the Hoxb7-cre;mT/mG mouse labels collecting duct cells, which can be identified by flow cytometry post-DISSECT (Fig. EV4A, A'). UUO-induced injury triggered proliferative responses to various degrees in different mice, which correlated with p-RB proliferative signaling (Fig. EV4B) (Giacinti & Giordano, 2006). Furthermore, after partial hepatectomy, BrdU-labelled hepatocytes (Fig. EV4C, D) were enriched for p-RB signaling during the recovery phase (Fig. EV4E). These results demonstrate DISSECT to be a valid, reliable approach for disaggregating a variety of heterogeneous epithelial tissues into single cell suspensions for cytometry-based signaling analysis.

DISSECT preserves signal transduction across a wide range of signaling pathways in epithelial tissues

We expect comparable quantitative approaches to have relatively comparable signal-to-noise detection. With regards to noise, we compared the standard deviation of signals generated from biological replicates using different quantitative approaches. Results generated by DISSECT followed by flow cytometry matched with those obtained by lysate-based ELISA assays and quantitative immunofluorescence imaging, demonstrating that these assays pick up comparable levels of noise (Appendix Fig. S6). With regards to signal, we performed rigorous, quantitative comparisons of TNF- α -induced signaling measurements between DISSECT-flow cytometry and two gold standard methods: quantitative immunofluorescence imaging (Fig. 3) and quantitative immunoblotting (Appendix Fig. S7). A summary of how we derived quantitative information from each of the three methods is documented in Appendix Figure S8. The same set of antibodies was used for all three methods to evaluate protein states, such as phosphorylation and cleavage, that act as direct surrogates of signaling pathway activation. Three cohorts of mice (30 samples) were used for each analysis, and tissues from each animal were split three ways for different types of analyses. Because lysate-based approaches assess

the average of all cell types in whole tissue, our cytometry analyses were also performed in a bulk cell population manner to enable direct comparison between approaches. To sample a wide dynamic range, we leveraged tissues from the duodenum and ileum (which exhibit differential TNF- α signaling responses), as well as from different time points post TNF- α exposure to generate quantitative correlation analyses. Ten out of eleven protein analytes generated statistically significant correlations between DISSECT-flow quantification and imaging quantification (6 out of 6 with quantitative immunoblotting) (Fig. 3, Appendix Fig. S7). Combined correlation analyses using all protein analytes resulted in a highly significant correlation ($p < 0.0001$) between DISSECT-flow and imaging data, and between DISSECT-flow and immunoblotting data. The Pearson's coefficients of comparing DISSECT-flow to imaging and immunoblotting were 0.72 and 0.81, respectively. Factors that contribute to the imperfect correlation include inherent experimental noise and differences in quantification between each of the methods, which will be discussed below. Furthermore, for a truly unbiased analysis, we did not exclude obvious data outliers that affected the normalization procedure, which can skew relatively small datasets and can subsequently weaken the correlations. Nevertheless, our conservative approach for validation still generated highly significant ($p < 0.0001$) correlations. These results demonstrate the validity of DISSECT to preserve native signaling during single cell disaggregation, and to generate single cell level data, when aggregated as populations, detect similar signal-to-noise as "gold standard" population-based methods.

DISSECT application of CyTOF identifies a differentially signaling enterocyte subpopulation that is sensitized to TNF- α -induced cell death

A 21-analyte CyTOF panel of heavy-metal labeled reagents specific for epithelial signaling was generated (Appendix Table S1). Twenty one-plex CyTOF analysis was performed on three cohorts of mice subjected to a time course of acute TNF- α exposure, giving rise to average early and late signaling results that matched with flow cytometry, imaging and

quantitative immunoblotting (Fig. 4A). We used single cell CyTOF data to first reaffirm TNF- α induction of cell death strictly within the duodenal enterocyte population. Indeed, CC3 did not colocalize with other epithelial cell type-specific markers (CK18: cytokeratin 18 – secretory subset, CLCA1 - goblet, CHGA- enteroendocrine, CD45 - leukocytes) (Fig. 4B, C compared to Fig. EV1E). The few double positive cells are not cell clusters (Appendix Fig. S9). The fraction of differentiated cell types detected again matched published results (Rojanapo *et al*, 1980; Paulus *et al*, 1993; Imajo *et al*, 2014; van der Flier & Clevers, 2009; Wright & Alison, 1984; Gunawardene *et al*, 2011; Cheng & Leblond, 1974; Gerbe *et al*, 2011), as well as flow and imaging data we obtained previously (Fig. 2B, 4B). To identify subpopulations of enterocytes with distinct signaling activities indicative of cell death, we used t-SNE (t-Distributed Stochastic Neighbor Embedding) to visualize multiplex single cell data in 2 dimensions while maintaining dissimilarities between cells in multidimensional data space (Fig. 4D, Dataset EV1) (Amir *et al*, 2013). We again focused on the one-hour time point to characterize active signaling cells undergoing cell death. t-SNE analysis allowed groupings of different functional cell types based on combinations of signaling and cell identity markers. In addition, a distinct population of CC3+ enterocytes was identified. We used manual gating on t-SNE space to supervise a partial least squares discriminant (PLSDA) model to categorize enterocytes undergoing cell death against living enterocytes. Classification based upon calibration signaling data in 2-latent variable PLSDA space to predict CC3 expression resulted in an area (AUC) of 0.92 under the receiver of operating characteristic (ROC) curve, indicative of high sensitivity and specificity (Fig. 4E). We then cross-validated our model by repeatedly withholding 10% of the data using random, venetian blind, and block selection. Our cross-validation model yielded similar prediction power (ROC AUC = 0.92) compared to our calibration model due to the high number of data points used for fitting a model with a relatively limited set of parameters, which dramatically lowers the prospects of overfitting. We used the discriminant coefficients (β) of our PLSDA model to select signaling features that were informative for classification. Using 10000-fold permutation testing,

we generated β distributions around zero and determined the probability for obtaining our model coefficients. The four coefficients with the lowest p-values were p-P38, p-CREB, p-ERK, and CK20 (Fig. 4F). Another method for feature selection using Variable Importance in Projection (VIP) scores also identified the same four variables (Fig. 4G). We overlaid these four variables onto t-SNE plots to determine their ability to predict CC3 expression (Fig. 4H). While individual variables positively or negatively correlated with the CC3+ population, they were incapable of clearly discerning this population from other cellular populations (Fig. 4I). Linearly combining these four variables without scaling allowed for clear identification of CC3+ enterocytes (Fig. 4J), indicating that combinatory activities of multiple signaling pathways contribute to a “signaling code” that implicate cell death. More importantly, the same experimental and computational analysis applied to three different cohorts of mice selected the same set of four variables that identify CC3+ enterocytes (Fig. 5, Dataset EV2, EV3). In addition, other β coefficients besides the top four variables also followed the same trend of positive or negative correlation with CC3 in different mouse cohorts. These results indicate that DISSECT followed by CyTOF is a highly reproducible method to accurately characterize single cell behavior using multi-pathway signaling parameters.

Divergently responding enterocytes are neighbors within the intestinal epithelium

Having a signaling fingerprint that classifies dying and non-dying enterocytes allows us to identify divergent signaling mechanisms that significantly affect intestinal physiology. Specifically, we chose to investigate divergent p-ERK signaling in the intestinal epithelium, which occurred in the surviving, but not in the dying, cell population. p-ERK activation in surviving enterocytes was also heterogeneous, which prompted us to envision spatial patterns of p-ERK activity that conferred survival. Whole-mount imaging of whole villus at 1 hour post TNF- α exposure revealed a “flower petal” ring-like pattern of epithelial p-ERK signaling, with five or six p-ERK positive cells surrounding a p-ERK negative area (Fig. 6 A, Fig. EV5A, yellow

arrows). Co-staining with CC3 revealed that in many cases, the dying CC3⁺ cells occupied the central area surrounded by p-ERK⁺ neighbors (Fig. 6B, Fig. EV5B, yellow arrows). In other cases, the dying CC3⁺ cell has already been extruded from the epithelium, leaving an apoptotic rosette surrounded by p-ERK⁺ cells undergoing contraction-dependent closure (red arrow). Furthermore, the ratios of CC3⁺ dying cells and p-ERK⁺ enterocytes in 3 cohorts of mice were 1:4.56, 1:6.04, 1:4.73, respectively, supporting that the immediate neighbors of the dying cell activated p-ERK signaling (Fig. EV5C, D). Imaging of tissue sections also corroborated that dying cells were flanked by p-ERK⁺ cells (Fig. EV5E), although the phenomenon was harder to visualize in 2 dimensions. We surmise that the dying cell signals to neighboring cells non-autonomously to activate a cell survival program, in order to prevent large swaths of contiguous epithelium from dying and to prevent unrecoverable barrier defects. Thus, we tested the effect inhibiting p-ERK signaling using the allosteric MEK inhibitor PD0325901 (Fig. EV5F). Inhibition of p-ERK signaling affected the latency of the cell survival program such that epithelial apoptosis occurred immediately following TNF- α exposure, which resulted in a higher number of dying cells in total (Fig. 6C). Inhibition of P38 alone minimally affected TNF- α induced apoptosis (Fig. EV5G), but was able to partially normalize early apoptosis due to MEK inhibition (Fig. 6C), consistent with P38's pro-apoptotic role that is context-dependent. To our knowledge, this is the first reported observation of this "flower petal" pattern of p-ERK activation in response to TNF- α -induced cell death in epithelial tissue. This new finding demonstrates the applicability of our single cell signaling experimental platform, in conjunction with data analysis, to reveal novel, non-cell autonomous responses in complex heterogeneous epithelia.

Discussion

A longstanding challenge for the expansion of multi-parameter cytometric analyses into epithelial signaling is the disruption of native signaling during single cell disaggregation. While techniques have been derived to detect epithelial structural proteins by single cell cytometric

approaches (Yamashita, 2007), activated signaling components have never been shown to be quantifiable. The DISSECT procedure precisely overcomes this limitation by preserving native signaling states in single epithelial cells. The quantitative yield of single cells recovered is demonstrated to be higher than that of conventional dissociation methods for cytometric applications. Application of multiplex single cell analyses enables the investigation of tissue heterogeneity that is characterized at the functional level by protein signaling. Natural variation of single cells, if accurately quantified, can be leveraged to generate tens of thousands of data points for building highly powered mathematical models. Our approach can reproducibly generate quantitative results, as supported by repeatable, robust conclusions drawn from mathematical modeling over multiple independent experiments in different animals. Furthermore, DISSECT has wide applications for either fluorescent flow cytometry or mass cytometry, and has demonstrated effectiveness in a broad range of epithelial tissues for interrogating *in vivo* signal transduction at single cell resolution or in a cell type-specific fashion.

Our approach to interrogate single cell signaling in epithelial tissues has several advantages over other single cell assays. Common single cell isolation approaches such as the Fluidigm C4 platform allow collection of only hundreds of cells, which limits the statistical power of downstream analyses (Trapnell *et al*, 2014). *In situ* approaches that require tissue sectioning result in inaccuracies in single cell quantification, since it is very difficult to control how much of each cell is retained during tissue sectioning. Specifically for intestinal epithelial cells that have diameters of ~ 10 to 40 μm (depending on the axis measurement), 5 μm tissue sections result in partial analyses of cells that contribute to measurement noise. Multiplex imaging techniques, either iterative (Gerdes *et al*, 2013) or heavy-metal based (Giesen *et al*, 2014; Angelo *et al*, 2014), are relatively low throughput and can take many hours of imaging for one sample, compared to 15 minutes per sample on the CyTOF. Arraying tissues on a slide can increase the throughput of imaging but at the expense of whole tissue sampling of large numbers of cells,

since a small region of the tissue will be the focus of a particular array core. Similarly, sectioning can only provide a very localized representation of whole tissue unless comprehensive serial sectioning and analysis are performed, a proposal only practical for small scale studies. However, compared to *in situ* methods such as RNA *in situ* hybridization (Itzkovitz *et al*, 2011), disaggregation into single cell suspensions eliminates all spatial context information. We can overcome this limitation by coupling cytometric analyses with imaging-based analyses such as MultiOmyx microscopy (Gerdes *et al*, 2013). Cell positions in cytometric analyses can then be cross-referenced to multiple markers characteristic of a cell's location determined by imaging. These marker-cell location relationships can be used for building "geographical maps", where independent cytometric datasets can be projected onto a spatial context.

Certain protein markers correlated much better than others when comparing the three experimental approaches. Partly, this is a reagent issue common in antibody-based detection assays, given that the access to a particular antigen is different under different denaturing and fixation conditions. Consequently, comparison between even the two traditional signaling analysis approaches, immunofluorescence imaging and immunoblotting, does not yield perfect correlation (Appendix Fig. S10). Our cytometric approach uses a different procedure to expose antigens, and is expected to exhibit some differences. Due to this reason, a specific advantage of our approach is its ability to detect a wider variety of antigens inaccessible to traditional immunohistochemistry. For example, the stem cell marker LRIG1 can only be detected in fresh or frozen tissues but not in paraffin embedded fixed tissues, but it is accessible to DISSECT-cytometry (Poulin *et al*, 2014).

Other sources of noise that can dampen the correlations include differences by which the average signal is quantified between the different methods (via median intensity in cytometry, the integrated intensity of an immunoblot band, and the mean pixel intensity in

imaging). Because of our limitations in defining intestinal cell borders in a confident manner using conventional imaging, we chose a highly reliable, unbiased way to establish nuclear and cytoplasmic masks for measuring signals in those subcellular compartments (Lau *et al*, 2007). This method, although simple, gives repeatable results especially with manual input, but comes with the added caveat that nuclear signals are fully represented whereas cytoplasmic signals represent a sampling of the perinuclear region. This may explain the sole discord in p-P90Rsk quantification given that this signal exists solely in the cytoplasm. Furthermore, quantification of tissue section images relies on microscope/camera-dependent pixel intensities in slivers of partial cells that are 5 μm thick, whereas flow cytometry quantifies the integrated voltage pulse generated by whole fluorescent particles. These differences in data generation were further amplified by our normalization procedure that can be affected by obvious outliers. Given these conditions, the high significance resulting from our correlation analyses is a testament to the robustness of DISSECT for generating quantitative results.

We previously published a model that selected features of TNF- α -induced cell death in the murine small intestine using lysate-based ELISA assays (Lau *et al*, 2011). Data variation was generated by examining different regions of the gut, which exhibit differential responses, or by using genetic mutations that affect TNF- α sensitivity (Lau *et al*, 2013, 2012). Our current approach to leverage natural variation in the same tissue can more accurately identify direct effectors of cellular behavior, since analyses of drastically different experimental contexts tend to select for secondary correlates. For instance, the duodenum and ileum are markedly distinct tissues (Bates *et al*, 2002), and features selected by modeling these variations may exaggerate the inherent differences between the tissues rather than actual modulators of TNF- α responses.

Our analysis identified that combinations of p-P38 and p-ERK MAP kinase pathway activities are critical determinants of TNF- α -induced cell death in the intestinal epithelium. A large body of literature over the past two decades has described P38 and ERK activation as

responses to TNF- α -induced inflammatory stress in epithelial cells (Bian *et al*, 2001; Ho *et al*, 2008; Jijon *et al*, 2005; Kim *et al*, 2005; Saez-Rodriguez *et al*, 2009; Song *et al*, 2003). In these bulk cell studies, p-P38 and p-ERK are implied to be co-regulated in the same cells as stress signals. P38 activation has been shown to be required for cell death downstream of TNF- α in a variety of contexts (Yu *et al*, 2014; Wu *et al*, 2015); co-activation of ERK by TNF- α has also been shown to be required (Qi *et al*, 2014). Our previous analyses of bulk lysate data also identified the MEK-ERK pathway to be positively correlated with cell death (Lau *et al*, 2011). However, our current results demonstrate that ERK is not activated in the relatively small fraction of dying enterocytes, but is activated heterogeneously in the remaining cells as a secondary response, resulting in its overall up-regulation at a whole tissue level. Activation of p-ERK occurs in direct neighbors surrounding the dying cell, forming a “flower petal” ring-like pattern. We propose that dying cells send signals to neighboring cells to activate a survival program, in order to prevent large swaths of neighboring cells from dying. Previous studies have demonstrated that an epithelial cell in the apoptotic process can signal to its neighbors to initiate purse-string contraction, generating enough force for cell extrusion (Rosenblatt *et al*, 2001; Monier *et al*, 2015). We reason that when more than one contiguous cell undergo apoptosis, contraction-dependent wound closure by surrounding cells becomes suboptimal, which leads to loss of epithelial integrity. Thus, epithelia have evolved intercellular communication programs for cells neighboring dying cells to survive. The molecular mechanisms responsible for this novel survival phenomenon remain to be identified, but may involve ATP released from the apoptotic cell, secretion of RTK ligands, or secondary responses downstream of cytoskeleton-dependent contraction (Patel *et al*, 2015; Boyd-Tressler *et al*, 2014; Kawamura *et al*, 2003; Xing *et al*, 2015). Consequently, inhibiting this survival mechanism both accelerated and increased TNF- α -induced cell death. Because of the complex *in vivo* regulation of intact epithelium, there are most likely other redundant, MEK-independent mechanisms in place to prevent wholesale cell death. Our study is distinct from other epithelial wound healing studies that focus on local cell

targeting (e.g., by laser ablation). Instead, divergent outcomes arise from epithelial cells exposed to the same apoptotic stimulus. This phenomena is also different from compensatory proliferation (Li *et al*, 2010), as cell death-driven proliferation in our system takes place in the crypt proliferative zone and not in the villus (Lau *et al*, 2011). Our novel epithelial cell-based CyTOF analysis allows us to identify heterogeneous signaling responses at the individual cell level with novel intercellular implications. Our epithelial application of cytometry-based technologies is useful for high-resolution dissection of heterogeneous responses in a complex tissue, and will have broad applicability to disease modeling, therapeutic design, and regenerative medicine.

Methods

Tissue Collection

Female C57BL6/J Mice (Jackson Laboratory) were administered 0.4mg/kg TNF- α in PBS via retro-orbital injection and sacrificed at time points ranging from 30 minutes and 4 hours post injection. Control mice were injected with PBS and sacrificed at 30 minutes post-injection. These mice (and their microbiomes) were acclimatized to Vanderbilt's mouse facility for at least 4 weeks. Upon sacrifice, 5 cm sections of duodenum and ileum were removed, washed using PBS, and spread longitudinally onto Whatman paper. Epithelial tissue was then separated from the muscle layers using a razor blade and transferred to a fixative solution of 4% paraformaldehyde (PFA) (Affymetrix) with protease (Roche) and phosphatase inhibitors (Sigma). After 30 minutes fixation at room temperature, tissues were washed twice in PBS and re-suspended in a solution of 1% BSA and 0.005% sodium azide in PBS for storage of up to 2 months. The number of animals used to generate data for each experimental time point/condition is indicated in Figure Legends, but most were of n=3.

As required, tissues from the same mice were either directly lysed in lysis (RIPA) buffer supplemented with protease and phosphatase inhibitors for immunoblotting analysis, or fixed in

4% PFA overnight for histological and imaging analysis. Histological samples were transferred to 70% ethanol and embedded in paraffin. Colonic tumors were collected from a $Lrig1^{Cre/+};Apc^{fl/+}$ tumor model (Powell *et al*, 2012), and they were cut into smaller fragments prior to processing. Colonic crypts were isolated from a standard EDTA chelation protocol (Sato *et al*, 2011).

Partial hepatectomy

All surgeries were performed according to the National Institutes of Health guidelines for the humane treatment of laboratory animals according to the “Guide for the Care and Use of Laboratory Animals” (NIH publication 86-23) and with approval of the Institutional Animal Care and Use Committee of Vanderbilt University Medical School. Prior to hepatectomy or for sham operations, mice were anesthetized with 60 mg/kg ketamine (Hospira) and 7mg/kg xylazine (Phoenix Pharmaceuticals) and positioned supine. A transverse incision was made inferior to the xiphoid process, which was excised. The median and left lateral lobes were eviscerated and ligated resulting in 60% liver removal. 5mg/kg/day of recombinant human IGF-1 (Cell Sciences) dissolved in water was given continuously with an osmotic pump (DURECT). To quantitate dividing hepatocytes, mice received one mg intraperitoneal injection of 5-Bromo-2-deoxyuridine (BrdU; BD Pharmingen) 2 hours before sacrifice. At tissue recovery, mice were anesthetized and weighed. Livers were excised, rinsed, blotted, and weighed. Sections were fixed in 10% neutral buffered formalin. Mortality after hepatectomy was <5% and not associated with a particular genotype.

Unilateral ureteral obstruction

Unilateral ureteral obstruction was performed, as previously published (Gewin *et al*, 2010), on mice aged 8-12 weeks by exposing the right kidney through a flank incision and ligating the ureter with two sutures just distal to the renal pelvis.

Perfusion of Fixative

Perfusion of animals was performed according to standard protocol (Gage *et al*, 2012) with 4% PFA with protease and phosphatase inhibitors. Perfusion was considered successful upon

observation of fixation tremors within one minute of perfusion. 20 mL of perfusion was performed, which was then followed by dissection, tissue collection, and standard DISSECT procedure as described.

Declaration of approval for animal experiments

All animal experiments were performed under protocols approved by the Vanderbilt University Animal Care and Use Committee and in accordance with NIH guidelines.

Conventional Disaggregation

Conventional disaggregation was performed on live epithelial tissue following the protocol by Magness *et al.* (Magness *et al.*, 2013). Disaggregation was also performed the same way without enzymatic digestion amidst a much lower yield of single cells. Single cells were then immediately fixed and permeabilized using a standard phospho-flow protocol (Kruzick *et al.*, 2004).

DISSECT Disaggregation

Epithelial tissues were re-suspended in -20°C acetone, then immediately pelleted and re-suspended in a detergent solution (1% Saponin, 0.05% Triton-x, 0.01% SDS) and agitated at room temperature for 30 min. Tissues were then blocked (2.5% donkey serum in PBS) at room temperature for 15 minutes before incubation with primary antibody. Samples were incubated in primary antibodies overnight at room temperature, and in secondary antibodies if needed for 1 hour at room temperature. Upon completion of antibody incubation, tissues were re-fixed in 4% PFA (Affymetrix). Disaggregation was then carried out with collagenase type I (Calbiotech) and dispase (Life Technologies) both at 1mg/mL in PBS, incubated at 37°C for 1 hour. Tissues were then gently and mechanically dissociated until most of the tissue has disaggregated into a cloudy solution. Samples were then washed and incubated for 5-10 minutes in nucleic acid intercalator (Hoescht/Iridium) to label the DNA, then filtered with a 45 µM meshed for cytometry (flow cytometry or CyTOF) analyses.

Cytometry Analyses

For both flow cytometry and CyTOF, cells were initially gated from debris using DNA content (Hoescht/Iridium). This was followed by size gating to eliminate cell clusters to obtain mostly cells with 2n/4n DNA content (Appendix Fig. S2). Single cells were then analyzed for intensities of antibody conjugates. Flow cytometry was performed on a BD LSRII with 5 lasers and CyTOF was performed on a Fluidigm-DVS CyTOF 1 instrument.

Quantitative Immunoblotting

Immunoblotting was performed using standard procedures and quantified using a LICOR Odyssey Fc imaging system. The top and bottom of the bounding box were used for background subtraction. Integrated intensity of immunoblot bands were taken after background subtraction (Appendix Fig. S8).

Quantitative Immunofluorescence Imaging

Mouse intestinal tissues were processed using standard immunohistological techniques and sectioned at 5 μm . Quantitative imaging was performed on an Olympus IX-81 inverted fluorescent microscope with a robotic stage for automated imaging of multiple fields of view. All images were first manually processed to eliminate stromal components. Automated image processing was then performed using custom ImageJ scripts. For determining cell fractions, masks were generated from the marker of interest and then quantified. Quantification was then normalized to the mask generated by nuclear staining with Hoescht. For co-localization, the intersecting mask from two sets of masks was obtained and then quantified as above. For signaling quantification, a nuclear mask was made from the nuclear channel and a 2 pixel thick cytoplasmic mask was made 5 pixels away from the nuclear mask. The target signal channel was quantified within the nuclear or cytoplasmic mask depending on if the signal was nuclear or cytoplasmic. The mean pixel intensity of the target signal was used for comparing between methods (Appendix Fig. S8). A single time course was stained, imaged and quantified per slide, and multiple technical replicates from serial section were performed. Villi lengths were measured by the pixel lengths from tips of villi to bases of crypts at 2x magnification.

Data Analysis

t-SNE analysis was performed using the viSNE implementation on Cytobank.org (Amir *et al*, 2013). Manual gating on t-SNE was performed by drawing contour lines based on density with 10% of the least dense data points excluded from the contours. The contours of the remaining cells represented cell populations grouped by their densities. PLSDA modeling, permutation testing, and feature selection were performed on Matlab (Mathworks). Unpaired t-tests were performed using Prism (Graphpad). Datasets EV1, EV2, EV3 can be accessed respectively at:

<https://www.cytobank.org/cytobank/experiments/48137>

<https://www.cytobank.org/cytobank/experiments/48139>

<https://www.cytobank.org/cytobank/experiments/48138>

Antibody Reagents

See Appendix Table S1. For optimization of antibodies for immunofluorescence imaging, a detailed procedure is documented in the Antibody Validation section in (Gerdes *et al*, 2013). Briefly, the procedure includes an antibody titration to determine the concentration range for optimal signal-to-noise detection. Further specificity testing included, but not limited to, immunogen peptide blocking, phosphatase treatment of samples to verify phosphospecificity, and visual inspection of expected localization patterns. The same antibody reagents were used across all experimental platforms. For optimization of antibodies for DISSECT-cytometry, similar titration studies were performed. Because the DISSECT procedure entails dissociation after staining, the localization of staining (crypt-villus/subcellular localization) was verified.

Competing Financial Interests

M.J.G. is an employee of General Electric. The authors declare no competing financial interests.

Acknowledgement

K.S.L is partially funded by a CCFA Career Development Award (308221), an Innovator Award from the AACR-Landon Foundation (15-20-27-LAUK), a career development award from the

Vanderbilt GI SPORE (P50-CA095103), VICTR (2UL1TR000445), and pilot project grants from the Vanderbilt-Ingram Cancer Center (P30-CA068485) and the Vanderbilt Digestive Disease Research Center (P30-DK058404). K.S.L is indebted to the VICC Young Ambassadors for their generous funding. C.A.H. is funded by the Training Program in Stem Cell and Regenerative Developmental Biology at Vanderbilt (T32-HD007502). R.J.C., J.L.F. and M.J.G. are partially funded by R01CA174377 from the NIH. R.J.C. is partially funded by the Vanderbilt GI SPORE (P50-CA095103). E.T.M. is funded by the Training Program in Basic and Translational Imaging of Cancer (R25-CA092043). J.M.I. is partially funded by R00CA143231. The authors would like to thank Kevin Weller and David Flaherty at the Vanderbilt Flow Cytometry core and Robert Carnahan at VAPR for their technical assistance.

Author Contributions

A.J.S. performed cytometry and imaging experiments. A.B. performed lysate experiments. E.T.M. assisted with imaging experiments. C.R.S. assisted with image analysis. C.A.H. performed computational modeling of the data. J.L.F. assisted with animal perfusion experiments and intellectually contributed. L.S.G. performed UUU experiments. R.M. and S.J.K. performed partial hepatectomy experiments. M.J.G., J.M.I., and R.J.C. intellectually contributed to the study and the writing of the manuscript. K.S.L. conceived of the study, initiated the development of the method, performed some of the cytometry and imaging experiments and imaging analysis, performed computational modeling of the data, wrote the manuscript, and supervised the research.

Figure Legends

Figure 1: The DISSECT disaggregation procedure enables cytometric analysis to investigate heterogeneous TNF- α signaling responses in tissue. (A) Representative immunofluorescence imaging (IF) of cells undergoing heterogeneous, position-independent

TNF- α -induced apoptosis in the villus as marked by CC3. (B) Non-overlapping localization between MUC2 (marking goblet cells) and CC3 at 1 hour post TNF- α administration. Extrusion of cells does not necessarily occur at the villus tips. (C) Expression of TNFR1 at basolateral cell membranes of villus epithelial cells in (C) vehicle-treated tissues and (C') loss of the receptor following TNF- α exposure. (D) Schematic of the DISSECT procedure for preserving native epithelial signaling during single cell isolation. Detergent solution is 1% Saponin, 0.05% Triton-x, 0.01% SDS.

Figure 2: DISSECT preserves phospho-protein signaling and cell identity marker expression. (A) IF of intact intestinal tissues compared to single cells prepared with DISSECT, stained for p-C-JUN early and p-STAT3 late in response to TNF- α . (A') Quantification of these single cell preparations by flow cytometry, with median values matching previous lysate-based results (Lau *et al*, 2011). (B) Flow cytometric quantification of epithelial cell identities following DISSECT, as determined by CLCA1 - goblet, CHGA - enteroendocrine, and DCLK1 – tuft cells at steady state. (B') Representative IF of these cell types on the same tissue, (B'') quantified by image analysis that match cytometric results. Error bars represent standard error of the mean (SEM) from n=8 fields of view. * $P \leq 0.05$, ** $P \leq 0.01$, *** $P \leq 0.001$, **** $P \leq 0.0001$

Figure 3: Quantitative comparison between single cell cytometric data and IF data of phospho-protein signaling markers. Quantification of single cells prepared from the intestinal epithelium using DISSECT followed by flow cytometry (solid lines) was compared to quantification of the same tissue by IF imaging analysis (broken lines). The dynamics of activation for each protein signaling marker by TNF- α from the duodenum and ileum were captured throughout a time course post TNF- α exposure (left column). Quantitative data from different time points and/or different regions were used to generate a range of variation for correlation analysis between DISSECT-Flow and IF for each signaling marker (right column). Error bars represent SEM from n=3 animals. A total of n=30 samples were used for each

correlation. Data scales are Z-score values derived from mean centering and variance scaling of each time course experiment (See Appendix Figure S8).

Figure 4: DISSECT disaggregation applied to CyTOF to investigate TNF- α signaling heterogeneity at single-cell resolution. (A) A sample of CyTOF signaling data generated from DISSECT in the intestinal epithelium as a TNF- α stimulation time course compared to other quantitative approaches. Data scales are normalized as in Figure 3. (B) CyTOF quantification of cells expressing villus epithelial cell markers only (CLCA1 – goblet cells, CK18 – subset of secretory cells, CHGA – enteroendocrine cells, CD45 –leukocytes), or their co-expression with CC3. Error bars represent SEM from n=3 animals. (C) Bi-plots generated by CyTOF demonstrating B for one sample. (D) t-SNE analysis of 21-dimensional single cell data demonstrating the segregation of cell types by signaling and cell identity marker expression (Dataset EV1). (E) The ROC curve of a 2-dimensional PLSDA model used for selecting features classifying enterocytes undergoing cell death against those that do not. Blue line represents the calibration model built with all data, while the green line represents the average of cross validation models built with partial data. (F) Determinant coefficients of the model with error bars representing the standard deviation around 0 over 10000 permuted runs. Stars denote the four most statistically significant coefficients. (G) VIP scores of the model, with scores greater than 1 representing importance in classification. (H) t-SNE map with heat representing CC3 expression, (I) p-P38, p-CREB, p-ERK1/2, CK20, and (J) combination of the four markers.

Figure 5: Analysis and modeling of 21-dimensional data over multiple biological replicates. For description, see Figure 4D-J. The same set of features were statistically identified to drive classification of enterocytes undergoing apoptosis over independent experiments (A, B) (Dataset EV2, EV3).

Figure 6: p-ERK activated in cells neighboring the dying cell promotes survival. (A) Whole villus imaging of p-ERK “flower petal” ring pattern surrounding a dying cell. (B) Examples

CC3+ cells surrounded directly by clusters of p-ERK+ neighbors (yellow arrows); an example of contraction-dependent closure by p-ERK+ cells after dying cell has been extruded (red arrow). (C) Flow cytometry of CC3+ cells induced by TNF- α under conditions of control, MEK inhibition and MEK and P38 inhibition. Quantified in C' with error bars representing SEM from n=3 animals. (D) Model of cell death-dependent activation of survival signaling in neighboring cells. Direct neighbors to the dying cell are instructed to survive to prevent contiguous patches of cell death unrecoverable by simple contraction-dependent closure.

Figure EV1: TNF- α induces heterogeneous villus epithelial cell death. (A) The fraction of CC3-positive cells identified by immunofluorescence (IF) image analysis over a time course of TNF- α induction. Error bars represent the standard error of the mean (SEM) from n=3 mice per time point. Cell death percentage integrated over time is 34.25 +/- 0.05 %. (B) Villi length quantified by image analysis, normalized to vehicle control. Error bars represent SEM from n=3 animals. (C) Morphology of villi 48 hours post TNF- α administration compared to vehicle control and 1 hour post TNF- α that exhibits severe villus blunting. (D) Representative IF of non-overlapping MUC2 and CC3 1hr post TNF- α administration. (E) IF quantification of cells expressing villus epithelial cell markers only (MUC2 – goblet cells, DCLK1 – tuft cells, CHGA – enteroendocrine cells), or their co-expression with CC3 1hr post TNF- α administration. Error bars represent SEM from n=3 animals. (F) IF of VIL co-localization with CC3 1hr post TNF- α administration. (G) Increased density of MUC2-positive cells detected by IF as cell death occurs. (H) Quantification of mature villus cell types over time by IF image analysis using stereotypic markers (MUC2 – goblet, DCLK1 – tuft, CHGA – enteroendocrine), for both duodenum and ileum. Error bars represent SEM from n=3 animals. (I) IF for expression of (I) TNFR1 and (I') TNFR2 in the villus. * P \leq 0.05, ** P \leq 0.01, *** P \leq 0.001, **** P \leq 0.0001

Figure EV2: DISSECT preserves native signal transduction during disaggregation. Comparison between early (p-ERK, p-C-JUN) and late (p-STAT3) signaling data generated from

quantitative immunoblotting, conventional disaggregation method followed by flow cytometry, and DISSECT followed by flow cytometry. Immunoblotting data represent integrated intensity of an immunoblot band. Flow cytometry data represents the median intensity of single cell distributions. Data scales are Z-score values derived from mean centering and variance scaling of each time course experiment (Appendix Figure S8). Error bars represent SEM from biological duplicates.

Figure EV3: DISSECT generates single cell suspensions with comparable yield to the conventional disaggregation method. (A) Whole field immunofluorescence image of single cell suspensions post-disaggregation. The conventional sample was imaged prior to methanol permeabilization. Autofluorescence was used to image the cell body. (B) Phase contrast image of single cell suspensions. (C) Quantitative comparison of yields between different disaggregation conditions starting from the same amount of intestinal material from the same animal. Error bars represent n=4 technical replicate. Two independent repeats was performed on different animals.

Figure EV4: DISSECT applied to other epithelial tissues also preserves intact signaling. (A) Collecting duct epithelial cells of the kidney marked by GFP (Hoxb7-Cre;mT/mG). (A') Single GFP+ collecting duct cells in A detected by flow cytometry post-DISSECT. (B) Proliferation of collecting duct cells 3 days post-unilateral ureteral obstruction (UUO) injury marked by Ki67, and proliferative p-RB signaling. Variation of proliferative response from different mice correlates with p-RB signaling. (C) Hepatocyte proliferation post-partial hepatectomy labelled by BrdU. (D) Single BrdU+ cells detected by flow cytometry post-DISSECT in control (D) and post-partial hepatectomy (D'). (E) p-RB proliferative signaling is enriched in BrdU+ proliferating population compared to non-proliferating BrdU- population.

Figure EV5: Divergent p-ERK signaling cells are neighbors to dying cells. (A) Distribution of p-ERK ring pattern along the whole villus. (B) More Examples CC3+ cells surrounded by

clusters of p-ERK⁺ neighbors (yellow arrows). (C) Gating of p-ERK⁺ cells from CyTOF data. A vehicle control (C) and a TNF- α treated sample (C') were used to gate for cells with homeostatic p-ERK levels versus activated p-ERK levels, respectively. (D) p-ERK⁺ cells and (D') dying cells plotted in t-SNE space for cohort 1. The percentages of dying and p-ERK⁺ cells were used to calculate ratio of dying to p-ERK⁺ cells. (E) Immunofluorescence of CC3 and p-ERK in duodenal tissue sections at 1 hour post TNF- α . CC3⁺ cells were flanked by p-ERK⁺ cells. (F) The efficacy of MEK inhibition assessed by p-ERK stimulation by TNF- α in the duodenum at 0.5 hour. (G) Percentage of CC3⁺ cells by flow cytometry in the context of P38 inhibition. Error bars represent SEM of biological duplicates.

References

- Abrahamsen I & Lorens JB (2013) Evaluating extracellular matrix influence on adherent cell signaling by cold trypsin phosphorylation-specific flow cytometry. *BMC Cell Biol.* **14**: 36
- Amir E-AD, Davis KL, Tadmor MD, Simonds EF, Levine JH, Bendall SC, Shenfeld DK, Krishnaswamy S, Nolan GP & Pe'er D (2013) viSNE enables visualization of high dimensional single-cell data and reveals phenotypic heterogeneity of leukemia. *Nat. Biotechnol.* **31**: 545–52
- Angelo M, Bendall SC, Finck R, Hale MB, Hitzman C, Borowsky AD, Levenson RM, Lowe JB, Liu SD, Zhao S, Natkunam Y & Nolan GP (2014) Multiplexed ion beam imaging of human breast tumors. *Nat. Med.* **20**: 436–42
- Bates MD, Erwin CR, Sanford LP, Wiginton D, Bezerra JA, Schatzman LC, Jegga AG, Ley-Ebert C, Williams SS, Steinbrecher KA, Warner BW, Cohen MB & Aronow BJ (2002) Novel genes and functional relationships in the adult mouse gastrointestinal tract identified by microarray analysis. *Gastroenterology* **122**: 1467–1482
- Baum B & Georgiou M (2011) Dynamics of adherens junctions in epithelial establishment, maintenance, and remodeling. *J. Cell Biol.* **192**: 907–917
- Bendall SC, Davis KL, Amir EAD, Tadmor MD, Simonds EF, Chen TJ, Shenfeld DK, Nolan GP & Pe'er D (2014) Single-cell trajectory detection uncovers progression and regulatory coordination in human B cell development. *Cell* **157**: 714–725
- Bendall SC, Simonds EF, Qiu P, Amir ED, Krutzik PO, Finck R, Bruggner R V, Melamed R, Trejo A, Ornatsky OI, Balderas RS, Plevritis SK, Sachs K, Pe'er D, Tanner SD & Nolan GP (2011) Single-cell mass cytometry of differential immune and drug responses across a human hematopoietic continuum. *Science* **332**: 687–96

- Bian ZM, Elnar SG, Yoshida a, Kunkel SL, Su J & Elnar VM (2001) Activation of p38, ERK1/2 and NIK pathways is required for IL-1beta and TNF-alpha-induced chemokine expression in human retinal pigment epithelial cells. *Exp. Eye Res.* **73**: 111–121
- Boyd-Tressler A, Penuela S, Laird DW & Dubyak GR (2014) Chemotherapeutic Drugs Induce ATP Release via Caspase-gated Pannexin-1 Channels and a Caspase/Pannexin-1-Independent Mechanism. *J. Biol. Chem.*
- Chaudhary R & Ghosh S (2005) Infliximab in refractory coeliac disease. *Eur. J. Gastroenterol. Hepatol.* **17**: 603–604
- Cheng H & Leblond CP (1974) Origin, differentiation and renewal of the four main epithelial cell types in the mouse small intestine. I. Columnar cell. *Am. J. Anat.* **141**: 461–79
- Colombel JF, Sandborn WJ, Reinisch W, Mantzaris GJ, Kornbluth A, Rachmilewitz D, Lichtiger S, D'Haens G, Diamond RH, Broussard DL, Tang KL, van der Woude CJ & Rutgeerts P (2010) Infliximab, azathioprine, or combination therapy for Crohn's disease. *N. Engl. J. Med.* **362**: 1383–1395
- Fatehullah A, Appleton PL & Näthke IS (2013) Cell and tissue polarity in the intestinal tract during tumourigenesis: cells still know the right way up, but tissue organization is lost. *Philos. Trans. R. Soc. Lond. B. Biol. Sci.*
- Van der Flier LG & Clevers H (2009) Stem cells, self-renewal, and differentiation in the intestinal epithelium. *Annu. Rev. Physiol.* **71**: 241–60
- Gage GJ, Kipke DR & Shain W (2012) Whole Animal Perfusion Fixation for Rodents. *J. Vis. Exp.*: 1–9
- Gerbe F, van Es JH, Makrini L, Brulin B, Mellitzer G, Robine S, Romagnolo B, Shroyer NF, Bourgaux J-F, Pignodel C, Clevers H & Jay P (2011) Distinct ATOH1 and Neurog3 requirements define tuft cells as a new secretory cell type in the intestinal epithelium. *J. Cell Biol.* **192**: 767–80
- Gerbe F, Legrauerend C & Jay P (2012) The intestinal epithelium tuft cells: specification and function. *Cell. Mol. Life Sci.* **69**: 2907–17
- Gerdes MJ, Sevinsky CJ, Sood A, Adak S, Bello MO, Bordwell A, Can A, Corwin A, Dinn S, Filkins RJ, Hollman D, Kamath V, Kaanumalle S, Kenny K, Larsen M, Lazare M, Li Q, Lowes C, McCulloch CC, McDonough E, et al (2013) Highly multiplexed single-cell analysis of formalin-fixed, paraffin-embedded cancer tissue. *Proc. Natl. Acad. Sci. U. S. A.* **110**: 11982–7
- Gewin L, Bulus N, Mernaugh G, Moeckel G, Harris RC, Moses HL, Pozzi A & Zent R (2010) TGF-beta receptor deletion in the renal collecting system exacerbates fibrosis. *J. Am. Soc. Nephrol.* **21**: 1334–1343
- Giacinti C & Giordano a (2006) RB and cell cycle progression. *Oncogene* **25**: 5220–5227

- Giesen C, Wang H a O, Schapiro D, Zivanovic N, Jacobs A, Hattendorf B, Schüffler PJ, Grolimund D, Buhmann JM, Brandt S, Varga Z, Wild PJ, Günther D & Bodenmiller B (2014) Highly multiplexed imaging of tumor tissues with subcellular resolution by mass cytometry. *Nat. Methods* **11**: 417–22
- Gunawardene AR, Corfe BM & Staton C a (2011) Classification and functions of enteroendocrine cells of the lower gastrointestinal tract. *Int. J. Exp. Pathol.* **92**: 219–231
- Halpern MD, Clark J a, Saunders T a, Doelle SM, Hosseini DM, Stagner AM & Dvorak B (2006) Reduction of experimental necrotizing enterocolitis with anti-TNF-alpha. *Am. J. Physiol. Gastrointest. Liver Physiol.* **290**: G757–64
- Ho AWY, Wong CK & Lam CWK (2008) Tumor necrosis factor-alpha up-regulates the expression of CCL2 and adhesion molecules of human proximal tubular epithelial cells through MAPK signaling pathways. *Immunobiology* **213**: 533–544
- Imajo M, Ebisuya M & Nishida E (2014) Dual role of YAP and TAZ in renewal of the intestinal epithelium. *Nat. Cell Biol.* **17**:
- Irish JM, Hovland R, Krutzik PO, Perez OD, Bruserud Ø, Gjertsen BT & Nolan GP (2004) Single cell profiling of potentiated phospho-protein networks in cancer cells. *Cell* **118**: 217–228
- Itzkovitz S, Lyubimova A, Blat IC, Maynard M, van Es J, Lees J, Jacks T, Clevers H & van Oudenaarden A (2011) Single-molecule transcript counting of stem-cell markers in the mouse intestine. *Nat. Cell Biol.* **14**: 106–114
- Jijon HB, Walker J, Hoentjen F, Diaz H, Ewaschuk J, Jobin C & Madsen KL (2005) Adenosine is a negative regulator of NF-κB and MAPK signaling in human intestinal epithelial cells. *Cell. Immunol.* **237**: 86–95
- Kawamura S, Miyamoto S & Brown JH (2003) Initiation and transduction of stretch-induced RhoA and Rac1 activation through caveolae. Cytoskeletal regulation of ERK translocation. *J. Biol. Chem.* **278**: 31111–31117
- Kim J-A, Kim D-K, Kang O-H, Choi Y-A, Park H-J, Choi S-C, Kim T-H, Yun K-J, Nah Y-H & Lee Y-M (2005) Inhibitory effect of luteolin on TNF-alpha-induced IL-8 production in human colon epithelial cells. *Int. Immunopharmacol.* **5**: 209–217
- Krutzik PO, Irish JM, Nolan GP & Perez OD (2004) Analysis of protein phosphorylation and cellular signaling events by flow cytometry: techniques and clinical applications. *Clin. Immunol.* **110**: 206–21
- Lau KS, Cortez-Retamozo V, Philips SR, Pittet MJ, Lauffenburger DA & Haigis KM (2012) Multi-scale in vivo systems analysis reveals the influence of immune cells on TNF-α-induced apoptosis in the intestinal epithelium. *PLoS Biol.* **10**: e1001393

- Lau KS, Juchheim AM, Cavaliere KR, Philips SR, Lauffenburger DA & Haigis KM (2011) In vivo systems analysis identifies spatial and temporal aspects of the modulation of TNF- α -induced apoptosis and proliferation by MAPKs. *Sci. Signal.* **4**: ra16
- Lau KS, Partridge E a., Grigorian A, Silvescu CI, Reinhold VN, Demetriou M & Dennis JW (2007) Complex N-Glycan Number and Degree of Branching Cooperate to Regulate Cell Proliferation and Differentiation. *Cell* **129**: 123–134
- Lau KS, Schrier SB, Gierut J, Lyons J, Lauffenburger DA & Haigis KM (2013) Network analysis of differential Ras isoform mutation effects on intestinal epithelial responses to TNF- α . *Integr. Biol. (Camb)*. **5**: 1355–65
- Li F, Huang Q, Chen J, Peng Y, Roop DR, Bedford JS & Li C-Y (2010) Apoptotic cells activate the 'phoenix rising' pathway to promote wound healing and tissue regeneration. *Sci. Signal.* **3**: ra13
- Lin CC, Huang WL, Su WP, Chen HHW, Lai WW, Yan JJ & Su WC (2010) Single cell phospho-specific flow cytometry can detect dynamic changes of phospho-Stat1 level in lung cancer cells. *Cytom. Part A* **77**: 1008–1019
- Magness ST, Puthoff BJ, Crissey MA, Dunn J, Henning SJ, Houchen C, Kaddis JS, Kuo CJ, Li L, Lynch J, Martin MG, May R, Niland JC, Olack B, Qian D, Stelzner M, Swain JR, Wang F, Wang J, Wang X, et al (2013) A multicenter study to standardize reporting and analyses of fluorescence-activated cell-sorted murine intestinal epithelial cells. *Am. J. Physiol. Gastrointest. Liver Physiol.* **305**: G542–51
- Monier B, Gettings M, Gay G, Mangeat T, Schott S, Guarner A & Suzanne M (2015) Apico-basal forces exerted by apoptotic cells drive epithelium folding. *Nature* **518**: 245–248
- O'Donnell EA, Ernst DN & Hingorani R (2013) Multiparameter Flow Cytometry : Advances in High Resolution Analysis. *Immune Netw.* **13**: 43–54
- Patel VA, Massenburg D, Vujicic S, Feng L, Tang M, Litbarg N, Antoni A, Rauch J, Lieberthal W & Levine JS (2015) Apoptotic cells activate AMPK and inhibit epithelial cell growth without change in intracellular energy stores. *J. Biol. Chem.:* jbc.M115.667345
- Paulus U, Loeffler M, Zeidler J, Owen G & Potten CS (1993) The differentiation and lineage development of goblet cells in the murine small intestinal crypt: experimental and modelling studies. *J. Cell Sci.* **106 (Pt 2)**: 473–483
- Pieters T, van Roy F & van Hengel J (2012) Functions of p120ctn isoforms in cell-cell adhesion and intracellular signaling. *Front. Biosci.* **17**: 1669
- Poulin EJ, Powell AE, Wang Y, Li Y, Franklin JL & Coffey RJ (2014) Using a new Lrig1 reporter mouse to assess differences between two Lrig1 antibodies in the intestine. *Stem Cell Res.* **13**: 422–430

- Powell AE, Wang Y, Li Y, Poulin EJ, Means AL, Washington MK, Higginbotham JN, Juchheim A, Prasad N, Levy SE, Guo Y, Shyr Y, Aronow BJ, Haigis KM, Franklin JL & Coffey RJ (2012) The pan-ErbB negative regulator Lrig1 is an intestinal stem cell marker that functions as a tumor suppressor. *Cell* **149**: 146–58
- Qi Z, Shen L, Zhou H, Jiang Y, Lan L, Luo L & Yin Z (2014) Phosphorylation of heat shock protein 27 antagonizes TNF- α induced HeLa cell apoptosis via regulating TAK1 ubiquitination and activation of p38 and ERK signaling. *Cell. Signal.* **26**: 1616–1625
- Rojanapo W, Lamb a J & Olson J a (1980) The prevalence, metabolism and migration of goblet cells in rat intestine following the induction of rapid, synchronous vitamin A deficiency. *J. Nutr.* **110**: 178–188
- Rosenblatt J, Raff MC & Cramer LP (2001) An epithelial cell destined for apoptosis signals its neighbors to extrude it by an actin- and myosin-dependent mechanism. *Curr. Biol.* **11**: 1847–1857
- Saez-Rodriguez J, Alexopoulos LG, Epperlein J, Samaga R, Lauffenburger DA, Klamt S & Sorger PK (2009) Discrete logic modelling as a means to link protein signalling networks with functional analysis of mammalian signal transduction. *Mol. Syst. Biol.* **5**: 1–19
- Sato T, van Es JH, Snippert HJ, Stange DE, Vries RG, van den Born M, Barker N, Shroyer NF, van de Wetering M & Clevers H (2011) Paneth cells constitute the niche for Lgr5 stem cells in intestinal crypts. *Nature* **469**: 415–8
- Schütze S, Tchikov V & Schneider-Brachert W (2008) Regulation of TNFR1 and CD95 signalling by receptor compartmentalization. *Nat. Rev. Mol. Cell Biol.* **9**: 655–662
- Song KS, Lee WJ, Chung KC, Koo JS, Yang EJ, Choi JY & Yoon JH (2003) Interleukin-1 β and tumor necrosis factor- α induce MUC5AC overexpression through a mechanism involving ERK/p38 mitogen-activated protein kinases-MSK1-CREB activation in human airway epithelial cells. *J. Biol. Chem.* **278**: 23243–23250
- Trapnell C, Cacchiarelli D, Grimsby J, Pokharel P, Li S, Morse M, Lennon NJ, Livak KJ, Mikkelsen TS & Rinn JL (2014) The dynamics and regulators of cell fate decisions are revealed by pseudotemporal ordering of single cells. *Nat. Biotechnol.* **32**: 381–6
- Williams JM, Duckworth C a, Watson AJM, Frey MR, Miguel JC, Burkitt MD, Sutton R, Hughes KR, Hall LJ, Caamaño JH, Campbell BJ & Pritchard DM (2013) A mouse model of pathological small intestinal epithelial cell apoptosis and shedding induced by systemic administration of lipopolysaccharide. *Dis. Model. Mech.* **6**: 1388–99
- Wright N & Alison M (1984) The biology of epithelial cell populations
- Wu H, Wang G, Li S, Zhang M, Li H & Wang K (2015) TNF- α -Mediated-p38-Dependent Signaling Pathway Contributes to Myocyte Apoptosis in Rats Subjected to Surgical Trauma. *Cell. Physiol. Biochem.* **150081**: 1454–1466

Xing Y, Su TT & Ruohola-baker H (2015) Tie-mediated signal from apoptotic cells protects stem cells in *Drosophila melanogaster*. *Nat. Commun.* **6**: 1–11

Yamashita S (2007) Heat-induced antigen retrieval: mechanisms and application to histochemistry. *Prog. Histochem. Cytochem.* **41**: 141–200

Yu L, Zhao Y, Xu S, Jin C, Wang M & Fu G (2014) Leptin confers protection against TNF- α -induced apoptosis in rat cardiomyocytes. *Biochem. Biophys. Res. Commun.* **455**: 126–132

Figure 1

Figure 2

p-C-JUN (S73) – Early Signal at 0.5 hr

p-STAT3 (Y705) – Late Signal at 2 hr

Figure 3

Signaling Data

Correlation

Signaling Data

Correlation

Signaling Data

Correlation

Legend

- Duodenum – DISSECT-Flow
- Ileum- DISSECT-Flow
- - - Duodenum – Imaging
- - - Ileum- Imaging

Figure 4

A

B

C

D

E

F

G

H

I

J

Figure 5

A

B

Figure 6

A

B

C

C'

D

Figure EV1

A

B

D

E

F

C

G

H

Figure EV2

Quantitative
Immunoblotting
(lysate)

Conventional
Method
(single cell)

DISSECT
(single cell)

Figure EV3

Conventional Disaggregation

DISSECT

A

B

C

Conventional (fix) – fix after conventional disaggregation
 Conventional (fix/spins) – add in same number of spins as DISSECT
 Conventional (fix/perm/spins) – add methanol permeabilization for phospho-flow

Figure EV4

Partial Hepatectomy (48h post-recovery)

E
Partial Hepatectomy
(48h post-recovery)

Figure EV5